# Carcass Traits, Meat Quality, and Volatile Compounds of Lamb Meat from Different Restricted Grazing Time and Indoor Supplementary Feeding Systems

**DOI:** 10.3390/foods10112822

**Published:** 2021-11-16

**Authors:** Bo Wang, Zhenzhen Wang, Yong Chen, Xueliang Liu, Kun Liu, Yingjun Zhang, Hailing Luo

**Affiliations:** 1State Key Laboratory of Animal Nutrition, College of Animal Science and Technology, China Agricultural University, Beijing 100193, China; wangboforehead@163.com (B.W.); wangzhenzhen198804@163.com (Z.W.); bravech@126.com (Y.C.); liuxueliang@cau.edu.cn (X.L.); liukun_139@163.com (K.L.); 2College of Animal Science and Veterinary Medicine, Heilongjiang Bayi Agricultural University, Daqing 163319, China; 3Henan Key Laboratory of Innovation and Utilization of Unconventional Feed Resources, Henan University of Animal Husbandry and Economy, Zhengzhou 450046, China; 4College of Grassland Science and Technology, China Agricultural University, Beijing 100193, China; zhangyj@cau.edu.cn

**Keywords:** meat characteristics, volatile compounds, flavor, feeding systems, lamb

## Abstract

This study was conducted to investigate the carcass traits, meat quality, and volatile compounds of growing lambs under different restricted grazing time and indoor supplementary feeding systems. Fifty 3-month-old male Tan lambs (with similar body weight) were divided into five groups randomly according to grazing time 12 h/d (G12), 8 h/d(G8), 4 h/d(G4), 2 h/d (G2), and 0 h (G0, indoor supplementary feeding). Animals were slaughtered at the end of the experiment, and the *longissimus thoracis* (*LT*) samples were collected for further analysis. The results indicated that indoor supplementary feeding improved the percentages of carcass fat and non-carcass fat of pre-slaughter weight (PSW) and decreased the cooking loss of lamb meat. Grazing for 8 h/d or 2 h/d enhanced PSW, carcass, and meat percentages of PSW. Lambs grazing for 2 h/d with supplement and indoor supplementary feeding lambs had a higher level of intramuscular fat and lightness (L*) value and lower cooking loss in the *LT* muscle, but higher yellowness (b*) and fat content were found in indoor supplementary feeding lambs. More categories of volatile compounds were identified in meat from grazing lambs than from indoor supplementary feeding lambs, but lower content of aldehydes and total volatile flavor compounds was detected in grazing lambs. Overall, the results demonstrated that the feeding system is a main factor that affects lamb meat quality, and proper grazing time can improve the quantity and quality of lamb meat and provide meat with different flavors to the consumers.

## 1. Introduction

China is the largest producer and consumer of mutton in the world with a production level of 4.92 million tons in 2020. In recent years, the demand for lamb meat has increased with economic development and the changing consumption preference of meat products. Meanwhile, consumers are paying more attention to meat quality due to the implications of meat composition for human health [1,2]. Many factors affect the organoleptic qualities and acceptability of lamb meat, such as sex, age, breed, and feeding systems [3,4,5]. Among these factors, feeding systems have been shown to play an important role in the regulation of meat quality. Current sheep feeding systems including outdoor grazing and indoor supplementary feeding, of which grazing is believed to be the cheapest method for lamb production and can also provide meat with the ‘green’ image as perceived by consumers [6]. It has been confirmed that grazing pastures can produce healthier lamb meat with higher accumulations of conjugated linoleic acids and *n*-3 polyunsaturated fatty acids and lower levels of saturated fatty acids and *n*-6/*n*-3 compared to meat from lambs that were fed a concentrate-based diet [7]. However, lambs under pure grazing management showed lower production performances compared to those that were fed a concentrate-based diet due to the low nutrient values in pastures [8]. Additionally, overgrazing results in a decrease in grassland productivity and damages the ecological balance, especially in desert or semidesert areas [9]. In contrast, supplemental feeding can improve the rate of live weight gain and produce heavier carcasses from the grazing livestock [10]. However, supplementation of commercial feed in the diet results in lower quality of lamb meat, such as higher contents of saturated fatty acids, cholesterol, and lower polyunsaturated fatty acids, improving the risk of an atherogenic potential in humans [11] and coronary heart disease [12]. Supplementation with concentrate could improve some indexes of meat quality such as the L* (lightness) value of meat [13,14]. These studies suggested that the lamb feeding systems face a contradiction between higher meat production with lower meat quality by indoor feeding and higher meat quality with lower production as well as the increase in environmental pressure by grazing.

A restricted grazing time system, which means giving the lamb constant grazing time on the grass ground per day with corresponding supplementary feed, can guarantee lambs both access to pastures and abundant energy requirements from supplemented concentrate. The effects of restricted grazing time systems on the pasture intake, productivity, and grazing behavior of dairy cows [15] and sheep [16] have been studied. Previous research has reported the effects of restricted grazing time systems on the foraging behavior, dry matter intake, weight gain, and other performances of growing lambs and grassland production [9,17,18]. The results indicated that moderate grazing time with the appropriate supplemental concentrate diet is a better way to manage lamb production when considering the growth performance of lambs and the productivity of the grasslands. In addition, our previous study showed that reasonable grazing with supplementary feeding can improve the fatty acid composition of lamb meat [19]. However, few reports have evaluated the influence of the restricted grazing system on the carcass traits and meat characteristics of growing lambs in a desert steppe.

Moreover, meat flavor is one of the important attributes of meat quality, plays a key role in influencing consumers’ acceptability of meat, and is closely associated with the volatile profile of meat [20,21]. Tan lamb is very popular among consumers because of its unique flavor and delicious meat quality. Tan lambs were traditionally raised in the desert and semidesert areas of Ningxia Hui Autonomous Region, in the northwest of China. However, limited studies have been undertaken on the relationship between meat flavor and feeding systems in Tan lamb. Therefore, the present study was conducted to study the effects of different restricted grazing systems on the carcass traits, meat quality, and volatile flavor compounds of growing Tan lambs in desert steppe and to provide an insight into the improvement of dietetic value and eating quality of lamb meat under different grazing time.

## 2. Materials and Methods

### 2.1. Animals and Experimental Design

This study was carried out in Yanchi County, Ningxia Hui Autonomous Region, China. The protocol used throughout the study was approved by the Institutional Animal Care and Use Committee of the China Agricultural University and in accordance with the Animal Ethics Committee of Beijing, China (DK996).

Fifty 3-month-old male Tan lambs (with similar body weights of 15.62 ± 0.33 kg) were selected from a local farm and divided into 5 groups (G12, G8, G4, G2, and G0) randomly. The experiment lambs were allocated different feeding strategies for 120 days prior to slaughter. A total of 52 hectares of grassland was fenced off into four equal areas, and lambs in each group of G12, G8, G4, and G2 were pastured on a 13-hectare natural grass ground for 12 h, 8 h, 4 h, and 2 h per day, respectively, while lambs in the G0 group were kept indoors and fed hay and concentrate. Different gradient commercial supplementary concentrate and hay were added to their diet. Detailed information of feeding strategies is presented in Table 1. The animals were housed in individual pens, except during grazing time, and were fed twice per day at 6:00 am before grazing and 7:00 pm after grazing. C32-alkane was used to estimate pasture intake of individual lambs, and the average dry matter intake of pasture in the G12, G8, G4, G2, and G0 groups was 1101.80 ± 126.87, 983.77 ± 97.62, 765.27 ± 46.59, 516.44 ± 47.64, and 668.09 ± 54.91 g/d, respectively [19]. All lambs had free access to water throughout the experiment.

The herbages of grassland were natural and mainly composed of three main herbages (Thymus mongolicus, Glycyrrhiza uralensis, and Caragana sinica) covering more than 80% of the land. Samples of each type of herbage were cut and collected manually at a height of 1 to 4 cm from 16 places in each plot at the beginning, middle, and end of the experimental period, respectively. All kinds of feed samples (three repetitions of each) including concentrate, herbages, and hay were analyzed for their chemical composition and the results are reported by their means in Table 2.

### 2.2. Slaughter Procedures and Carcass Measurements

After 120 days of the feeding experiment, the final body weight in the G12, G8, G4, G2, and G0 groups was 26.22 ± 0.81, 30.45 ± 0.94, 27.80 ± 1.17, 31.06 ± 0.96, and 27.42 ± 0.85 kg, respectively; the corresponding average daily gain was 101.35 ± 5.61, 157.16 ± 7.88, 127.76 ± 7.40, 157.02 ± 4.78, and 118.65 ± 6.05 g, respectively [19]. Before being slaughtered, the lambs were prevented from consuming feed for 24 h and from drinking for 8 h, and the pre-slaughter weight (PSW) was recorded. The lambs were slaughtered using halal practices according to the specifications of the Animal Care and Use Committee of China Agricultural University. After slaughter, the dressed carcasses (the weight without the skin, head, feet, and internal organs) were weighed (tailed hot carcass weight in the G12, G8, G4, G2, and G0 groups was 10.21 ± 0.35, 12.55 ± 0.58, 11.93 ± 0.55, 14.11 ± 0.60, and 9.96 ± 0.47 kg, respectively [19]). Omental and mesenteric fat, kidney fat, and tail fat were separated and weighed to calculate the percentage of PSW. The loin eye area was recorded using a planimeter on the cut surface of *longissimus thoracis* (*LT*) at the interface between the 12th and 13th rib on both sides of the carcass. The carcass was then split along the vertebral column into left and right halves [22]. The composition (meat, subcutaneous fat, and bone content) of the right half was carried out by dissection (conditions of the dissection room: temperature, 4 °C; air speed, 0.5 m/s) and percent distribution was calculated on the basis of PSW [23]. The left half was separated into leg, loin, rib, neck, shoulder, and breast to calculate their percentage of carcass.

### 2.3. Meat Quality Determination

After slaughter, *LT* muscle samples from the right half were collected to measure color parameters, pH, and cooking loss. Meat color was measured with a Minolta chromameter (CR-400, Minolta Inc., Osaka, Japan) after a 30 min blooming period on the section of loin eye area and expressed as CIE Lab lightness (L*), redness (a*), and yellowness (b*). Muscle pH was determined using a digital pH meter (Testo 205, Testo AG, Lenzkirch, Germany) equipped with a penetrating electrode and thermometer at 45 min and 24 h after slaughter on the surface of loin eye area and was expressed as pH_45min_ and pH_24h_. In order to measure cooking loss (%), meat samples (chilled at 4 °C for 24 h) were firstly weighed, then placed in plastic bags, and cooked in a water bath at 75 ℃ for 1 h as described by Hoffman et al. [24]. Bags including cooked samples were cooled under running cold water for 1 h, samples were dried with paper towels, and reweighed. Cooking loss (%) was estimated by means of percentage of weight loss of the cooked sample to the initial sample weight.

About 20 g *LT* muscle (1 day of ageing at 4 °C) from the right side of each carcass was sampled and dried using the freeze method [25] for 72 h until the sample weight was constant to determine the contents of moisture, fat, protein, and ash. The dry matter (DM), crude protein (CP), intramuscular fat (IMF), and ash of the meat were analyzed according to the Association of Official Analytical Chemists procedures [26]. Another 10 g of *LT* muscle was sampled in vacuum packages and frozen at −20 °C for volatile analysis.

### 2.4. Volatile Compounds Analysis

Five grams of each *LT* muscle sample were weighted and minced and then transported to a 10 mL headspace extraction bottle after thawing. The bottle was sealed after adding 2 g sodium chloride (analytically pure) and 1 μL 0.1632 μg/μL 2-methyl-3-heptanone and fully mixed, followed by a constant temperature water bath at 90 °C for 1 h. Volatile compounds were isolated using a solid-phase microextraction (SPME) needle consisting of fused silica fiber with 50/30 μm thickness and 2 cm length of a DVB/CAR/PDMS (divinylbenzene/carboxen/polydimethylsiloxane) layer (Supelco, Bellefonte, PA, USA). The pre-prepared SPME needle (heated in the GC injection port at 270 °C for 1 h before the extraction) was put above the sample headspace for 40 min at 37 °C by a water bath to achieve equilibrium. The equilibrium time was calculated according to Steffen and Pawliszyn [27]. The SPME fiber was desorbed directly at 250 °C for 5 min (2 min in splitless mode followed by 3 min with a purge flow of 60 mL/min) in the injection port of the GC and then prepared for the GC-MS analysis. After desorption of each sample, the fiber was cleaned for 2 min at 270 °C before the next extraction.

The volatile compounds were measured using a gas chromatograph-mass spectrograph (GC-MS, QP2010 Plus, Shimadzu, Japan). An HP-INNOWax capillary column (Agilent, Santa Clara, CA, USA) of 30 m × 0.32 mm I.D. × 0.25 μm film thickness was used for the separation of analytes. The setup of gas chromatography was as follows: Carrier: helium (5 mL/min); Pressure: 117.6 KPa; Column flow rate: 1.00 mL/min; Flow speed: 30.00 mL/min; and Temperature of inlet and interface: 250 °C. The procedure of temperature increase was: initial temperature was first set as 50 °C and kept for 1 min, then increased at a rate of 3 °C/min to 210 °C, followed by a 10 °C/min increase to 250 °C, and held for 18 min. The transfer line to the mass selective detector was kept at 150 °C. The setup of mass spectrograph was: Temperature of ion source: 230 °C; Quadrupole temperature: 150 °C; Ionizing voltage: EI (70 eV, 200 μA); and Scan range: 30 to 400 (m/z).

Volatile compounds were identified by the comparison of their mass spectra with the NIST 08 L Mass Spectral Database and the retention indices with those of known compounds. The linear retention index of each peak was calculated using a C6–C24 saturated alkane standard (Supelco, Bellefonte, PA, USA) as the same GC temperature program. The approximate quantities of the volatile compounds were estimated by comparing their peak areas with the n-alkane internal standard obtained from the total ion chromatogram. The results wer expressed as means ± standard error of all the repetitions in each group.

### 2.5. Statistical Analysis

The one-way analysis of variance was conducted by using the SAS 8.2 version (SAS 8.2, Institute Inc., Cary, NC, USA) to analyze the effect of different restricted grazing time and indoor supplementary feeding systems on carcass traits, meat quality, and volatile compounds. Significant differences among the G12, G8, G4, G2, and G0 groups were determined by Duncan’s post-hoc multiple comparisons test. Results were expressed as means ± standard error. Statistical differences were considered to be significant at *p* < 0.05.

## 3. Results

### 3.1. Carcass Quality

The PSW and carcass traits of Tan lambs are shown in Table 3. Lambs from the five treatments had different PSW values, although they had the same initial body weights. Lambs in the G2 group had the largest PSW, whereas lambs in the groups of G12 and G0 had the lowest PSW (*p* < 0.05). Compared with the G12 and G0 groups, higher values of carcass and meat percentages of PSW and loin eye area were observed in lambs of the G8, G4, and G2 groups (*p* < 0.05). The bone percentages in the G12, G8, and G4 groups were higher than that in the G0 group (*p* < 0.05). The accumulations of carcass (subcutaneous fat and tail fat) or non-carcass (omental and mesenteric fat and kidney fat) adipose tissues, which were described as the percentage of PSW and the value of meat/bone, in the G2 and G0 groups was higher than those in the G4 group, with the lowest in the G12 group. There were no obvious differences found in percentages of different standardized joints of the carcass (including shoulder, rib, loin, leg, and neck) (*p* > 0.05) among the groups, but breast percentage in the G12 group was significantly higher than that in the G4, G2, and G0 groups (*p* < 0.05).

### 3.2. Meat Composition and Quality

Results of the *LT* muscle chemical composition indicated a significant difference under different restricted grazing time and indoor supplementary feeding systems. Table 4 showed that the contents of crude protein and IMF in *LT* muscle were affected by different feeding systems (*p* < 0.05). The *LT* muscle of lambs in the G0 group had lower crude protein content than the other four groups (*p* < 0.05), among which there were no significant differences (*p* > 0.05). Higher IMF contents were detected in the muscle of G8, G2, and G0 groups than that of G12 group (*p* < 0.05), but there were no obvious differences among the G8, G4, G2, and G0 groups (*p* > 0.05). Additionally, the moisture and ash contests of the *LT* samples did not change significantly with the variation of grazing time and concentrate (*p* > 0.05).

The effects of restricted grazing time and indoor supplementary feeding systems on pH, color parameters, and cooking loss of *LT* muscle from Tan lambs are shown in Table 5. The pH_45min_ of *LT* muscle showed a nonsignificant decreasing trend as restricted grazing time increased (*p* > 0.05), and no significant differences were observed in the pH_24h_ of *LT* muscle (*p* > 0.05). The L* value of the *LT* muscle was the lowest in animals of the G12 group, which was significantly lower than those in the groups of G4, G2, and G0 (*p* < 0.05). Lambs from the G4 and G2 groups indicated higher a* values than those in the G8 group and lower b* values than those in the G8 and G0 groups (*p* < 0.05). As the amount of time spent grazing decreased and the amount of concentrate consumed increased (from G12 to G0), the levels of cooking loss in *LT* muscle gradually decreased with reduced grazing time (*p* < 0.05).

### 3.3. Volatile Flavor Compounds

The number of aldehydes identified in the G12, G8, G4, G2, and G0 groups was 12, 12, 11, 9, and 9, respectively (Table 6). The content of pentanal, hexanal, heptanal, 2-octenal, and 2-nonanal in the G0 group were significantly higher than in other groups (*p* < 0.05), but no differences were evident among the G12, G8, G4, and G2 groups (*p* > 0.05). Benzaldehyde had a higher concentration in the G0, G12, G8, and G4 groups than in the G2 group (*p* < 0.05). Tetradecanal was higher in the G12 group than in the G8 and G4 groups (*p* < 0.05) but was not detected in the G2 and G0 groups.

The content of alcohols was affected by different grazing time and indoor feeding systems (Table 7). There were seven alcohols detected in both the G12 and G8 groups, and 6, 5, and 3 alcohols were identified in the G4, G2, and G0 groups, respectively. Menthol content was higher in the G12 group than in the G8, G4, and G2 groups (*p* < 0.05) but was not detected in the G0 group. The contents of 1-octene-3-ol and 2-octenol were significantly higher in the G12 and G8 groups than in the G0 group (*p* < 0.05). Decanol was found in the G12, G8, G4, and G2 groups but was absent in the G0 group.

Table 8 shows the ketones, esters, and heterocyclic compounds of *LT* muscle affected by restricted grazing time systems in Tan lamb. No ketones were detected in the G0 group, and no esters were identified in both the G2 and G0 groups. 2-phenylethyl caproate was only found in the G12 and G8 groups and was present at a greater concentration in the G12 group (*p* < 0.05). The concentration of 2-pentylfuran was higher in the G0 group than the other groups (*p* < 0.05).

In the comparison of total content of different volatile flavor compounds among the groups, we found that five different chemical classes were detected in the G12, G8, and G4 groups, but four and three kinds were identified in the G2 and G0 groups, respectively (Table 9). The concentrations of aldehydes, heterocyclic compounds, and total volatile flavor compounds (TVFC) were the highest in the G0 group, and the lowest in the G2 group (*p* < 0.05). The alcohols, ketones, and esters decreased with the reduction in grazing time from the G12 group to the G0 group (*p* < 0.05).

## 4. Discussion

### 4.1. Carcass Traits

Feeding regime and nutritional level are important factors influencing production performances of sheep. In general, concentrate could provide higher energy than herbage feeds. Grazing itself is usually not able to satisfy the nutritional requirements of the grazing animal for best production performances due to the low-energy of pastures, although grazing the livestock on pasture could decrease production costs and improve profitability [28]. Concentrate supplements changed growth performance, live weights, carcass weight, meat production, bone weights, and back fat of growing lambs compared with grazing lambs [6,29]. These studies are accordance with our previous findings that lambs from the grazing with concentrate supplementation groups also indicated higher final body weight and average daily gain than those from the purely grazing lambs [19]. In the present study, changes in PSW of Tan lambs showed the same tendency, lambs in the G8 and G2 groups achieved higher PSW, which was probably due to the fact that animals in these groups received concentrate supplement with large amount of fresh herbages to meet the nutritional needs of growing lambs [19]. The final body weight, daily weight gain, pre-slaughter weight, and carcass weight were changed with similar patterns. As for the animals in the indoor feeding group, lower PSW might be due to the low quality hay (e.g., lower crude protein in the hay) in their diets and relatively lower pasture intake, which was similar to lambs grazed for 2 h/d [19]. This may also explain the results of loin eye areas and meat percentages as higher values were found in lambs from the G8, G4, and G2 groups; however, other studies have found that lambs fed with pastures had higher meat percentages [30]. High quality feeds (containing higher energy and protein) resulted in more meat accumulations and large loin eye areas of lambs in the G8, G4, and G2 groups. The feeding regime can also change the composition of animal bodies. Lambs from the G12 and G0 groups had lower carcass percentages of PSW than in the G8, G4, and G2 groups in this study; this may be due to the increase in digestive tract percentage such as heavier small intestine and higher rumen volume, which was stimulated by purely grazing and the indoor lambs fed with poor-quality hay ad lib [31]. In the present study, the meat to bone ratios increased with the decrease in lamb grazing time and the increase in concentrate supplementation, which was in accordance with previous findings [32]. A possible explanation is that grazing lambs had physical activity during grazing, which could stimulate bone growth [17,33].

Feeding systems are an important factor determining the carcass and non-carcass fat. Normally, lambs from a concentrate based system have a higher fat percentage, while lambs grazed at pasture have lower fat than stall-fed lambs [34,35]. This might be due to greater energy intake of lambs in feeding systems based on concentrate diet compared to those of grazing based systems [36]. In the current study, lambs in the G8, G4, G2, and G0 groups, which had access to concentrate, had a higher accumulation of subcutaneous fat, omental and mesenteric fat, kidney fat, and tail fat than the purely grazing group. Additionally, movement of grazing lambs could also influence the accumulations of fat, since increased physical activity causes an increase in mobilization of reserve lipids in order to form muscle tissue and subsequently reduce the fatness [34]. Therefore, the decreased grazing time and increased concentrate supplementation with higher energy intake tend to produce fatty mutton.

Among the standardized joints from left half carcasses, leg, shoulder, and ribs were the more important joints in the five treatments, whereas neck, loin and breast were of lesser importance; these results are in agreement with previous reports [37,38]. Normally, when studying effects on carcass components of concentrate- or grazing-fed lambs slaughtered at heavy weight, wholesale cuts related to motor functions, such as legs and shoulders, had greater percentages of carcass in forage-fed lambs due to greater muscular and bone development [35,38]. Therefore, other parts such as breast and neck had relatively lower percentages of the carcass. However, in our study, restricted grazing time systems had no effects on all joints except the part of breast. This phenomenon did not reflect the moving distance differences among the treatments, which showed that total moving distance of Tan lambs significantly increased with the increasing restricted grazing time [17] and would require further analysis to determine the impact.

### 4.2. Meat Composition and Quality

Moisture content is the largest component in lamb meat. Researchers reported that different feeding regimes did not affect moisture in mutton [39,40], similar to this study. Protein is the crucial nutrient in lamb meat, but its content in lamb meat was not affected by grazing or concentrate treatment [41]. The crude protein content in the meat of Tan lambs grazing on pasture was higher than in animals receiving drylot feeding in the present study. This variation might be due to the greater activity of grazing lambs [17], which prompted the decomposition of carbohydrate and fat and enhanced the synthesis of protein. The lambs in the G8, G2, and G0 groups received more concentrate and had relatively higher IMF contents than those in the G12 group. Nuernberg et al. [42] and Arvizu et al. [43] observed the same situation that drylot feeding with high concentration feeds could increase fat deposition in muscles. Notably, intramuscular fat, as one of the main attributes to evaluate meat quality, was correlated with a tendency to lower water-holding capacity, but better palatability, juiciness, tenderness, and higher flavor precursors with an increase in the fat content [44,45], and lower IMF indicated an adverse effect on eating satisfaction [46]. Moreover, lambs with proper grazing and concentrate supplementation feeding improved the fatty acid profile and indicated lower saturated fatty acid, higher *n*-3 polyunsaturated fatty acids, and ratio of *n*-3/*n*-6 compared to the indoor feeding [19]. Ash from meat is an important source of mineral for humans. In our study, the levels of ash from different treatments did not change significantly, which might be due to the fact that the lambs fed indoor had free access to the hay, and the ash level of the hay (8.75% of DM) was similar to the levels of some main herbages.

Meat quality involves a number of parameters such as palatability, color, nutritional value, and safety, among others. It can be affected by many factors including genetics, feeding systems, and processing conditions [47]. Under different production regimes, diet composition and feed availability can directly or indirectly influence meat quality [48]. Muscle pH_45min_ reflects the pH measured immediately in the meat postmortem [49]. Ultimate pH reflects the pH of muscle excised 24 h post mortem, and it is probably the most important indicator used to measure meat quality at a commercial level [50]. In the present study, the pH_45min_ and pH_24h_ of *LT* muscle in Tan lambs recorded did not change as grazing time and the amount of concentrate supplementation changed. The variation in ultimate pH among different feeding systems might be attributed to the variation in muscle glycogen contents, which was lower in lambs finished on pasture than lambs finished on a grain-based feedlot [51]. However, the effects of feeding regimes on muscle pH were inconsistent. Some results demonstrated that being fed with a commercial diet resulted in lower ultimate pH values when compared to grazing animals in reindeer [13] and lambs [31,41]. Some other studies reported nonsignificant differences in meat pH among different feeding systems [38,51].

Meat color is one of the main sensory quality parameters and is the most important factor that influences the consumers’ initial selection and purchase decision on meat or meat products [52]. Fresh meat of *longissimus thoracis et lumborum* (meat color was measured after a blooming period of 30–40 min) with a lightness value above 44 was acceptable by 95% of consumers, and below 34 was unacceptably dark for consumers [53]. The lightness value of *LT* muscle improved with dietary concentrate supplementation in growing Tan lambs in our study. Similarly, a previous study also reported a higher meat lightness in lambs raised on concentrate-based systems compared with grazed pasture counterparts [36]. The differences in meat color among grazing systems in the current study may be attributed to combined effects of carcass fatness level, intramuscular fat, and ultimate meat pH [54]. Moreover, the fattier carcasses had a slower cooling rate in the muscle and therefore a faster pH decline, and the difference in meat lightness between stall- and pasture raised lambs could be partially caused by the difference in the ultimate pH since higher pH meats tend to have a darker color [31]. However, the highest yellowness value presented in the G8 and G0 groups, which was inconsistent with our expectation that the value would increase with the increased pasture intake. The possible explanation was that it was related to factors such as diet composition and physical activities, which needs further research.

Cooking loss refers to the loss of meat during cooking, and it is determined by factors including genetics, feeding regime, and meat processing [46]. Cooking loss is one of the parameters to evaluate meat water holding capacity and is closely related to the juiciness of meat [55]. There were no differences in the cooking loss between pasture-fed and alfalfa/linseed pellet-fed lambs in a previous report [39], whereas in our study, as the concentrate in diets increased, the cooking loss of lambs’ *LT* muscle decreased. These differences may be explained as low cooking losses attributed to high intramuscular fat content of meat, since the high fat had a bundling effect on water during cooking [44]. In addition, the range of cooking loss value in lamb meat varies greatly in different studies, some reports range from 30% to 38% [23,56,57], and some from 20% to 27% [58,59,60]. The reasons for this difference have not been reported clearly. We speculate that it may be related to factors such as breed, age, slaughter weight, and diet composition, but further research is needed. 

### 4.3. Volatile Flavor Compounds

Lamb meat is very popular with many consumers because of its unique flavor. The flavor of lamb meat depends on the chemical composition of volatile (aroma) and nonvolatile (taste) compounds, and the quantity and balance of flavor molecules are critical to the acceptability of meat flavor [61]. Volatile compounds such as aldehydes, alcohols, ketones, phenols and others have been implicated as responsible for or contributing to lamb meat flavor [62,63]. Factors affecting volatile flavor compounds mainly include the genetics, age, sex, and cooking methods, especially diet [64]. A number of researchers have studied the volatile differences between pasture- and grain-based feeding systems, and a considerable number of volatile compounds were identified and reported associated with lamb meat flavor or can be used to discriminate between animals from different feeding systems [20,61,65,66].

In the current study, 28 volatile compounds were identified, and the odor descriptors were described according to previous studies [5,67,68]. Aldehydes were the highest percentage of volatile compounds and indicated a relatively lower odor threshold, which were considered as a critical composition of volatile flavor [69]. The indoor feeding lambs had a higher concentration of saturated aliphatic aldehydes such as pentanal, hexanal, heptanal, octanal, and benzaldehyde, but the concentration decreased with the combination of grass feeding and concentrate feeding. Higher concentration of these aldehydes may increase the undesirable odor production [21]. The alcohols were also an important member of volatile flavor with a higher odor threshold than aldehydes and relatively lower contribution to flavor [70]. Alcohols such as 1-octene-3-ol (green, mushroom, earthy, oily) and 2-octenol (green, fatty, citrus) decreased with grazing time limit in this study, which contributed to the lamb meat characteristics. Ketones have an obvious milk or fruit flavor with a relatively higher odor threshold but were not found in the indoor feeding lambs. Moreover, esters indicated limited contribution to meat flavor because of the relatively higher odor threshold and were not detected in the G2 and G0 groups. Additionally, most of the heterocyclic compounds were derived from complex reaction processes, such as lipid oxidation and Maillard reaction [71]. 2-pentylfuran which has a roast and buttery aroma showed a higher concentration in lamb meat from pure grazing or indoor feeding than in pasture combined with concentrate feeding. These results suggested that the categories of volatile decreased with the limited access of lambs to pasture grass, but the quantity of aldehydes and heterocyclic compounds increased in the indoor feeding lambs.

However, the influence of feeding systems on volatile compounds were inconsistent among different studies. Some studies reported that a concentrate-based diet could increase the levels of aldehydes, ketones, and lactones and provide a more intense mutton aroma [72,73]. On the other hand, the increased content of aldehydes, ketones, phenols, and indoles were reported based on the grazing system by previously research [65,74]. Moreover, no differences in lamb meat flavor attributes (taste and odor/aroma) were found when the lambs were fed with different diets [75]. Additionally, the different extraction methods used in these studies also indicated a huge influence on the volatiles that were recovered as each technique has a bias towards certain chemical classes. The mechanisms by which the type of feeding systems or diets can affect the lamb meat flavor are complex and multiple. In its simplest form, dietary composition may affect the final flavor of lamb meat by direct transfer of specific plant-derived compounds into the meat or they may undergo degradation during thermal processing to form new flavor-active compounds [68]. Interestingly, the acceptability of meat flavor is not only decided by the content of volatile compounds, but also nonvolatile and other factors influence consumer’s preference [76], which suggests the volatile flavor compounds evaluation should connect with a sensory test in future research to provide a more comprehensive assessment of meat flavor.

## 5. Conclusions

Restricted grazing time and indoor supplementary feeding systems in desert steppe affected the carcass and meat characteristics of growing Tan lambs. Lambs grazed for 2 h/d and lambs fed indoors received greater amounts of concentrates, indicated higher L* value, lower cooking loss, and IMF content, which has a positive relationship with meat quality, while the dissectible carcass fat (subcutaneous fat and tail fat) and non-carcass fat (omental and mesenteric fat and kidney fat) also increased in the indoor feeding lambs. Meat flavor was affected by different feeding systems via increasing the content of aldehydes, heterocyclic compounds, and total volatile flavor compounds in indoor feeding lambs but decreased the categories of flavor chemical classes. Additionally, the quantity of volatile flavor compounds decreased with the limited grazing time and increased concentrate supplementation. Consequently, feeding with more structured grazing time and concentrate supplementation could improve the meat quality of growing Tan lambs, and provide meat with different flavors.

## Figures and Tables

**Table 1 foods-10-02822-t001:** Different feeding strategies of the five groups [19].

Groups	Duration of Grazing (h/d)	Supplementary Concentrate Levels		
		First Two Months (g/d)	Last Two Months (g/d)	Total(kg)
G12	12	Non-supplemented	Non-supplemented	0
G8	8	150	300	27
G4	4	150	300	27
G2	2	300	500	48
G0	0 (feedlot)	300 And hay (full dose)	500 And hay (full dose)	48

**Table 2 foods-10-02822-t002:** Ingredients and chemical composition of concentrate, herbages, and hay [19].

Items	Concentrate	Herbages			Hay
		Thymus Mongolicus	Glycyrrhiza Uralensis	Caragana Sinica	
Ingredients, %					
Corn	59.5				
Wheat bran	10.0				
Soybean meal	25.5				
Premix ^1^	5.0				
Chemical composition, % of DM					
DM	87.5	40.89	33.88	30.79	85.99
CP	19.10	9.00	18.81	21.93	10.33
Ash	3.20	12.10	23.40	6.92	8.75
NDF	13.50	43.83	34.10	32.94	50.70
ADF	6.00	39.10	31.71	32.03	37.42

^1^ The premix per kilogram contains the following material: vitamin A (retinyl acetate), 120,000 IU; vitamin D_3_, 18,000 IU; vitamin E (DL-α-tocopheryl acetate), 500 IU; Fe, 900 mg; Cu, 150 mg; Mn, 1160 mg; Zn, 1900 mg; I, 11,000 mg; Se, 6 mg; Co, 6 mg; Ca, 100 g; P, 30 g; and NaCl, 150 g. DM, dry matter; CP, crude protein; NDF, neutral detergent fiber; and ADF, acid detergent fiber.

**Table 3 foods-10-02822-t003:** Carcass traits of Tan lambs under different restricted grazing time and indoor supplementary feeding systems.

Items	Treatments				
	G12	G8	G4	G2	G0
PSW (kg)	24.15 ± 0.76 ^c^	28.44 ± 0.91 ^ab^	25.85 ± 1.08 ^bc^	29.98 ± 1.04 ^a^	25.32 ± 0.74 ^c^
Meat/bone	2.10 ± 0.10 ^c^	2.23 ± 0.06 ^bc^	2.35 ± 0.16 ^bc^	2.73 ± 0.13 ^ab^	2.95 ± 0.36 ^a^
Loin eye area (cm^2^)	8.88 ± 0.42 ^b^	9.86 ± 0.51 ^ab^	10.24 ± 0.30 ^ab^	10.78 ± 0.60 ^a^	8.96 ± 0.41 ^b^
Percentage of (%) PSW					
Carcass (%)	42.27 ± 0.54 ^b^	45.42 ± 0.66 ^a^	46.09 ± 0.44 ^a^	46.99 ± 0.79 ^a^	40.42 ± 1.26 ^b^
Meat (%)	29.16 ± 0.48 ^d^	31.43 ± 0.43 ^b^	31.08 ± 0.49 ^bc^	34.76 ± 0.54 ^a^	29.70 ± 0.46 ^cd^
Bone (%)	12.68 ± 0.35 ^a^	13.36 ± 0.26 ^a^	13.24 ± 0.31 ^a^	11.75 ± 0.20 ^ab^	10.08 ± 0.22 ^b^
Subcutaneous fat (%)	1.26 ± 0.05 ^b^	1.94 ± 0.16 ^a^	1.47 ± 0.09 ^b^	2.23 ± 0.11 ^a^	2.09 ± 0.12 ^a^
Omental and mesenteric fat (%)	0.24 ± 0.02 ^b^	0.33 ± 0.05 ^ab^	0.29 ± 0.04 ^b^	0.32 ± 0.03 ^ab^	0.36 ± 0.04 ^a^
Kidney fat (%)	0.20 ± 0.03 ^b^	0.36 ± 0.04 ^a^	0.31 ± 0.02 ^a^	0.37 ± 0.03 ^a^	0.34 ± 0.02 ^a^
Tail fat (%)	1.85 ± 0.21 ^b^	2.27 ± 0.20 ^ab^	2.03 ± 0.21 ^b^	2.84 ± 0.16 ^a^	2.76 ± 0.27 ^a^
Percentage (%) of half carcass					
Shoulder (%)	22.68 ± 0.67	25.00 ± 1.51	24.28 ± 0.55	25.09 ± 0.67	25.22 ± 1.21
Rib (%)	11.64 ± 0.35	12.05 ± 0.78	11.99 ± 0.55	11.41 ± 0.48	12.41 ± 0.64
Loin (%)	10.53 ± 0.31	11.55 ± 0.80	11.14 ± 0.39	10.93 ± 0.47	11.05 ± 0.78
Breast (%)	4.52 ± 0.25 ^a^	4.30 ± 0.31 ^ab^	3.57 ± 0.18 ^c^	3.77 ± 0.19 ^bc^	3.55 ± 0.23 ^c^
Leg (%)	31.87 ± 0.53	33.28 ± 2.39	32.29 ± 0.74	32.17 ± 0.75	32.94 ± 1.30
Neck (%)	6.91 ± 0.91	6.79 ± 0.57	6.21 ± 0.55	7.23 ± 0.56	5.30 ± 0.56

Different superscripts (a, b, c and d) mean significant differences (*p* < 0.05). PSW, pre-slaughter weight. The values are the means ± SE (*n* = 10 per group). G12, G8, G4, and G2 mean that the lamb grazing time was 12 h/d, 8 h/d, 4 h/d, and 2 h/d, respectively. G0 means indoor supplementary feeding.

**Table 4 foods-10-02822-t004:** Chemical composition of *LT* muscle in Tan lambs under different restricted grazing time and indoor supplementary feeding systems.

Items	Treatments				
	G12	G8	G4	G2	G0
Moisture (%)	76.17 ± 0.16	75.08 ± 0.43	75.17 ± 0.44	75.45 ± 0.20	75.90 ± 0.46
Protein (%)	19.75 ± 0.14 ^a^	20.05 ± 0.15 ^a^	19.51 ± 0.19 ^a^	19.74 ± 0.33 ^a^	18.88 ± 0.21 ^b^
IMF (%) [19]	2.00 ± 0.18 ^b^	3.93 ± 0.63 ^a^	2.97 ± 0.43 ^ab^	3.81 ± 0.33 ^a^	3.62 ± 0.52 ^a^
Ash (%)	1.12 ± 0.06	1.03 ± 0.04	1.13 ± 0.03	1.19 ± 0.06	1.21 ± 0.05

Different superscripts (a and b) mean significant differences (*p* < 0.05). IMF, intramuscular fat. The values are the means ± SE (*n* = 10 per group). G12, G8, G4, and G2 mean that the lamb grazing time was 12 h/d, 8 h/d, 4 h/d, and 2 h/d, respectively. G0 means indoor supplementary feeding.

**Table 5 foods-10-02822-t005:** PH, meat color, and cooking loss of *LT* muscle of Tan lambs under different restricted grazing time and indoor supplementary feeding systems.

Items	Treatments				
	G12	G8	G4	G2	G0
pH_45min_	6.28 ± 0.06	6.27 ± 0.09	6.36 ± 0.08	6.46 ± 0.09	6.50 ± 0.06
pH_24h_	5.91 ± 0.04	5.91 ± 0.04	5.97 ± 0.04	5.91 ± 0.06	6.06 ± 0.06
L*	40.23 ± 1.33 ^a^	42.13 ± 1.22 ^ab^	43.58 ± 1.18 ^b^	44.82 ± 0.62 ^b^	44.80 ± 1.08 ^b^
a*	22.46 ± 0.69 ^ab^	21.43 ± 0.52 ^b^	24.22 ± 0.86 ^a^	23.56 ± 0.81 ^a^	22.02 ± 0.68 ^ab^
b*	12.42 ± 0.65 ^b^	14.65 ± 0.95 ^a^	12.73 ± 0.80 ^b^	12.72 ± 0.63 ^b^	14.17 ± 0.45 ^a^
Cooking loss (%)	39.71 ± 0.63 ^b^	38.58 ± 0.91 ^ab^	38.12 ± 0.66 ^ab^	37.48 ± 0.91 ^ab^	36.59 ± 0.55 ^a^

Different superscripts (a and b) mean significant differences (*p* < 0.05). The values are the means ± SE (*n* = 10 per group). L*, lightness. a*, redness. b*, yellowness. G12, G8, G4, and G2 mean that the lamb grazing time was 12 h, 8 h, 4 h, and 2 h per day, respectively. G0 means indoor supplementary feeding.

**Table 6 foods-10-02822-t006:** Influence of restricted grazing time and indoor supplementary feeding systems on the aldehydes of *LT* muscle in Tan lamb (ng/g).

Items	Treatments					Odor Descriptors
	G12	G8	G4	G2	G0	
Pentanal	2.13 ± 0.14 ^b^	2.25 ± 0.21 ^b^	1.93 ± 0.10 ^b^	1.91 ± 0.16 ^b^	5.35 ± 0.23 ^a^	Fermented, fruity, nutty, pungent
Hexanal	9.59 ± 0.45 ^b^	10.51 ± 1.04 ^b^	10.35 ± 0.99 ^b^	7.25 ± 0.68 ^b^	47.19 ± 3.12 ^a^	Herbal, grassy
Heptanal	8.73 ± 0.52 ^c^	12.17 ± 0.88 ^b^	7.50 ± 0.46 ^c^	7.56 ± 0.51 ^c^	23.63 ± 1.39 ^a^	Green, fresh, fatty, sweet
Octanal	19.83 ± 0.97 ^b^	14.42 ± 1.01 ^c^	21.94 ± 1.61 ^ab^	9.34 ± 0.85 ^d^	26.58 ± 2.68 ^a^	Citrus, floral
Nonanal	23.84 ± 3.19	26.13 ± 2.86	33.24 ± 2.07	36.23 ± 3.98	26.58 ± 3.74	Waxy, aldehydic, green, fresh
Benzaldehyde	46.97 ± 2.86 ^ab^	54.50 ± 4.78 ^ab^	42.23 ± 3.98 ^b^	25.77 ± 2.86 ^c^	59.27 ± 4.17 ^a^	Fruity, strong, sharp
2-ethylhexanal			1.05 ± 0.42	2.31 ± 0.10	3.01 ± 0.69	Rosy, sweet
2-octenal	2.15 ± 0.02 ^b^	1.96 ± 0.01 ^b^	1.82 ± 0.02 ^b^	2.04 ± 0.03 ^b^	4.65 ± 0.31 ^a^	Nutty, meaty
Decanal	2.05 ± 0.10	2.05 ± 0.24	4.29 ± 0.79			Aldehydic, sweet, waxy, green
Hexadecaldehyde	2.72 ± 0.12	2.98 ± 0.17				-
Tetradecanal	3.29 ± 0.20 ^a^	1.55 ± 0.10 ^b^	1.76 ± 0.05 ^b^			-
2-nonenal	2.12 ± 0.07 ^b^	1.80 ± 0.08 ^b^	2.00 ± 0.05 ^b^	1.88 ± 0.05 ^b^	4.95 ± 0.16 ^a^	Fatty, green, aldehydic
2,4-decadienal	1.48 ± 0.07	1.58 ± 0.04				Waxy, fatty, earthy, green

Different superscripts (a, b, c and d) indicate significant differences (*p* < 0.05). The values are the means ± SE (*n* = 10 per group). G12, G8, G4, and G2 mean that the lamb grazing time was 12 h, 8 h, 4 h, and 2 h per day, respectively. G0 means indoor supplementary feeding.

**Table 7 foods-10-02822-t007:** Influence of restricted grazing time and indoor supplementary feeding systems on alcohols content of *LT* muscle in Tan lamb (ng/g).

Items	Treatments					Odor Descriptors
	G12	G8	G4	G2	G0	
Menthol	2.10 ± 0.10 ^a^	1.63 ± 0.04 ^ab^	0.98 ± 0.01 ^b^	0.87 ± 0.01 ^b^		Alcoholic
Dodecanol	2.67 ± 0.10	1.99 ± 0.06				-
1-octene-3-ol	5.82 ± 0.17 ^a^	5.33 ± 0.41 ^a^	3.24 ± 0.29 ^b^	3.96 ± 0.12 ^ab^	2.64 ± 0.21 ^b^	Green, mushroom, earthy, oily
2-octenol	5.86 ± 0.29 ^a^	4.03 ± 0.32 ^ab^	3.82 ± 0.23 ^b^	3.43 ± 0.40 ^b^	1.82 ± 0.20 ^c^	Green, fatty, citrus
Decanol	2.54 ± 0.31	1.89 ± 0.12	1.79 ± 0.09	1.92 ± 0.11		-
Octadecanol	1.87 ± 0.07	1.65 ± 0.04	0.97 ± 0.01			-
Octanol	3.87 ± 0.10	3.24 ± 0.16	3.58 ± 0.20	4.01 ± 0.31	4.27 ± 0.29	Green, waxy, fruity

Different superscripts (a, b, and c) indicate significant differences (*p* < 0.05). The values are the means ± SE (*n* = 10 per group). G12, G8, G4, and G2 mean that the lamb grazing time was 12 h, 8 h, 4 h, and 2 h per day, respectively. G0 means indoor supplementary feeding.

**Table 8 foods-10-02822-t008:** Influence of restricted grazing time and indoor supplementary feeding systems on ketones, esters, and heterocyclic compounds of *LT* muscle in Tan lamb (ng/g).

Items	Treatments					Odor Descriptors
	G12	G8	G4	G2	G0	
Ketones						
Pentanone	2.79 ± 0.19	2.26 ± 0.10				Cheese
2-methylacetophenone	2.48 ± 0.14	2.16 ± 0.18	2.20 ± 0.16	1.98 ± 0.06		-
Esters						
2-phenylethyl caproate	16.15 ± 1.02 ^a^	8.74 ± 0.99 ^b^				-
Vinyl caproate	5.13 ± 1.00	4.97 ± 0.89	4.61 ± 0.91			-
2-Ethylhexyl acrylate	2.65 ± 0.17	2.33 ± 0.20	2.27 ± 0.21			-
Heterocyclic compounds						
2-ethylfuran	2.38 ± 0.08	3.85 ± 0.27	4.00 ± 0.65	2.96 ± 0.10	3.77 ± 0.24	Chemical, sweet, burnt, earthy
2-pentylfuran	14.65 ± 1.75 ^b^	4.50 ± 0.67 ^c^	4.82 ± 0.54 ^c^	6.54 ± 0.73 ^c^	25.24 ± 2.31 ^a^	Roast, buttery
3-methylthiophene	3.24 ± 0.42					-

Different superscripts (a, b, and c) indicate significant differences (*p* < 0.05). The values are the means ± SE (*n* = 10 per group). G12, G8, G4, and G2 mean that the lamb grazing time was 12 h, 8 h, 4 h, and 2 h per day, respectively. G0 means indoor supplementary feeding.

**Table 9 foods-10-02822-t009:** Influence of restricted grazing time and indoor supplementary feeding systems on the total aldehydes, alcohols, ketones, esters, and heterocyclic compounds of *LT* muscle in Tan lamb (ng/g).

Items	Treatments				
	G12	G8	G4	G2	G0
Aldehydes	124.9 ± 12.66 ^b^	131.9 ± 13.61 ^b^	128.11 ± 12.02 ^b^	94.29 ± 8.73 ^c^	201.21 ± 13.89 ^a^
Alcohols	24.73 ± 1.99 ^a^	19.76 ± 2.09 ^ab^	14.38 ± 1.81 ^b^	14.19 ± 1.52 ^b^	8.73 ± 1.08 ^c^
Ketones	5.27 ± 1.05 ^a^	4.42 ± 0.98 ^a^	2.20 ± 0.16 ^b^	1.98 ± 0.06 ^b^	
Esters	23.93 ± 2.10 ^a^	16.04 ± 1.93 ^b^	6.88 ± 1.18 ^c^		
Heterocyclic compounds	19.27 ± 2.09 ^b^	8.35 ± 1.68 ^c^	8.82 ± 2.01 ^c^	9.50 ± 1.27 ^c^	29.01 ± 3.11 ^a^
TVFC	198.1 ± 18.64 ^b^	180.47 ± 19.10 ^b^	160.39 ± 15.31 ^bc^	122.17 ± 12.05 ^c^	238.95 ± 21.75 ^a^

Different superscripts (a, b, and c) mean significant differences (*p* < 0.05). TVFC, total volatile flavor compounds. The values are the means ± SE (*n* = 10 per group). G12, G8, G4, and G2 mean that the lamb grazing time was 12 h, 8 h, 4 h, and 2 h per day, respectively. G0 means indoor supplementary feeding.

## Data Availability

Not applicable.

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
