# Peer review of "Carcass Traits, Meat Quality, and Volatile Compounds of Lamb Meat from Different Restricted Grazing Time and Indoor Supplementary Feeding Systems"

_foods, 2021, doi:10.3390/foods10112822_

Round 1

Reviewer 1 Report

The experimental plan and the interpretation of the results was good. I only found some minor points, which i have listed below.

Line 15: change conduct to conducted

Line 24: input lightness with your L value in the abstract, similar to the way you have discussed b value.

Line 41: change to . Current sheep feeding ......including outdoor....

Line 65: change to . Previous research has reported....

Line 138: change to. Bags including cooked.....

Line 171: please include interface temperature

Line 382: change to.  .....researches have studied...

You did not discuss the B value in your discussion of the meat color. I think your results for B value were not as expected as you would imagine that increased pasture should equate to increased b-carotene and therefore an increase in b value. You need to discuss this aspect.

In line 394 (reference 62) you state that alcohols have a lower odour threshold than aldehydes.  In general terms i don't think that statement is correct, typically aldehydes have a lower odour threshold than alcohols but it can be volatile specific, thus i would double check this statement.

In the section lines 407 to 413, one thing that you should mention is that all of these studies used different extraction methods and this has a huge influence on the volatiles that are recovered as each technique has bias towards certain chemical classes not to mention the different temperature, extraction times, etc.

I also think that if sensory analysis was included in the study, it would have been a huge benefit, although i realize that its unlikely at this stage, but i felt i should mention it for future studies.

Query re stats on volatiles, should zero values not be included for those samples that did not contain the volatile and if so should these not be included in the statistical evaluation?

Author Response

Reviewer 1

Dear reviewer,

We gratefully appreciate the constructive comments and suggestions; the manuscript has been revised according to your comments. Changes of the text are indicated in red.

The experimental plan and the interpretation of the results was good. I only found some minor points, which I have listed below.

Line 15: change conduct to conducted

Response to the reviewer: “conduct” was replaced by “conducted”.

Line 24: input lightness with your L value in the abstract, similar to the way you have discussed b value.

Response to the reviewer: “lightness” was supplemented in the revised manuscript.

Line 41: change to . Current sheep feeding ......including outdoor....

Response to the reviewer: The sentence was revised as suggested by the reviewer.

Line 65: change to . Previous research has reported....

Response to the reviewer: The sentence was revised according to the reviewer’s suggestion.

Line 138: change to. Bags including cooked.....

Response to the reviewer: The sentence was revised.

Line 171: please include interface temperature

Response to the reviewer: The information was supplemented in the revised manuscript.

Line 382: change to. .....researches have studied...

Response to the reviewer: The sentence was revised according to the reviewer’s comment.

You did not discuss the B value in your discussion of the meat color. I think your results for B value were not as expected as you would imagine that increased pasture should equate to increased b-carotene and therefore an increase in b value. You need to discuss this aspect.

Response to the reviewer: We discussed the yellowness value as suggestion by the reviewer.

In line 394 (reference 62) you state that alcohols have a lower odour threshold than aldehydes. In general terms I don't think that statement is correct, typically aldehydes have a lower odour threshold than alcohols but it can be volatile specific, thus I would double check this statement.

Response to the reviewer: The sentence was corrected by checking the reference again.

In the section lines 407 to 413, one thing that you should mention is that all of these studies used different extraction methods and this has a huge influence on the volatiles that are recovered as each technique has bias towards certain chemical classes not to mention the different temperature, extraction times, etc.

Response to the reviewer: Thanks for your suggestion. The discussion was supplemented according to your comments.

I also think that if sensory analysis was included in the study, it would have been a huge benefit, although I realize that its unlikely at this stage, but I felt I should mention it for future studies.

Response to the reviewer: Thanks for the reviewer’s suggestion, we will carry out sensory evaluation in the next research.

Query re stats on volatiles, should zero values not be included for those samples that did not contain the volatile and if so should these not be included in the statistical evaluation?

Response to the reviewer: Zero values were included in the statistical evaluation, and we supplemented the expression “Result was expressed as means ± standard error of all the repetitions in each group.” in the revised manuscript.

Reviewer 2 Report

While the data presented are of interest, there are serious issues with the study design and description of the results that should be solved or at least discussed appropriately. Thus, the manuscript should be revised substantially.

The authors present here a study that is obviously an extension of a former publication (Wang et al. Asian-Austral. J. Anim. 2015), but without referring to this publication and discussing it properly. Table 1 and 2 of the current manuscript have already been published in the former paper and must be referenced accordingly. Furthermore, IMF content has also been published before. The results of both studies should be discussed together and clarified what was the aim of the former study compared to the current study.

The discussion mainly considers the grazing time as the varied factor. However, the results are based on grazing time and different concentrate feeding levels, resulting in different energy intake. More factors should be considered and discussed, as movement, feed intake and energy intake have at least influenced the results. Thus, the study design enables and requires other comparisons than actually done, e.g. hey versus pasture feeding in addition to the same amount of concentrate (G2 vs. G0). Insights from other parts of the study should be included.

Specific comments:

L72 What is meant with “lambs synthetically”? Please reword to clarify.                   

Table 1 Remove from the manuscript or indicate that it was published before. Obviously, the different feeding strategies resulted in different energy intake among the groups, which should have the main influence on the results.  

L101 How was the pasture feed intake determined and used in later analyses? I do not see it considered in the results or discussion.

L118 What was the reason for slaughter lambs using halal procedures?

L120 Please clarify how you defined the EBW. What about life end weight, feed intake, energy intake and parameters of growth like daily weight gain? Those data should be provided or referenced, when published elsewhere. Can you show growth curves?

L164 correct “Shimadzu”

L180-183 Which are the fixed factors and random factors included in the statistical model?

Tables: include a footnote explaining the treatment groups. Insert in footnotes the superscript letters used to indicate significant differences.

Table 9: superscript letter “d” in G2 for TVFC should probably be “c”. If Heterocyclic compounds are specified in a footnote, this should be done for all classes of compounds or for none.

L273 How was the value of the provided feed in this study in comparison to the nutritional requirements of lambs? Include respective discussion.

L278 Discuss the relationship between live end weight and EBW for lambs in this study.

L306-307 This would be expected with increased energy intake.

L328-329 Lambs of the G4 group received the same amount of concentrate as G8 lambs according to Table 1.

L346-347 Clarify “restricted access time”

L375-378 Reword to clarify the sentence.

L382 delete “were”

L398-400 revise this sentence

L402 Do you mean “more intense” or “higher”?

L408 replace “concentrated-based“ with “concentrate-based”

 L413 replace “feed” with “were fed” and “mechanism” with “mechanisms”

L424 It is important to mention that these lambs received greater amounts of concentrates.

The Conclusion is mainly a summary and should be revised.

Author Response

Reviewer 2

Dear Reviewer,

Thank you for helping us improve the quality of manuscript (ID: foods-1451533). We gratefully appreciate the constructive comments made by you. We revised the manuscript according to your comments and suggestions. Any changes of the manuscript were indicated in red in the revised manuscript and listed as follows.

While the data presented are of interest, there are serious issues with the study design and description of the results that should be solved or at least discussed appropriately. Thus, the manuscript should be revised substantially.

The authors present here a study that is obviously an extension of a former publication (Wang et al. Asian-Austral. J. Anim. 2015), but without referring to this publication and discussing it properly.

Response to the reviewer: Information of previous publication was supplemented and discussed in the revised manuscript according to the reviewer’s suggestion.

Table 1 and 2 of the current manuscript have already been published in the former paper and must be referenced accordingly. Furthermore, IMF content has also been published before. The results of both studies should be discussed together and clarified what was the aim of the former study compared to the current study.

Response to the reviewer: The previously published data presented in this manuscript were cited and discussed as suggested by the reviewer.

The discussion mainly considers the grazing time as the varied factor. However, the results are based on grazing time and different concentrate feeding levels, resulting in different energy intake. More factors should be considered and discussed, as movement, feed intake and energy intake have at least influenced the results. Thus, the study design enables and requires other comparisons than actually done, e.g. hey versus pasture feeding in addition to the same amount of concentrate (G2 vs. G0). Insights from other parts of the study should be included.

Response to the reviewer: Thanks for the reviewer’s comments. Feeding system is the main factor considered in this study. Different feeding system including many factors such as feed intake, movement, energy intake and other parameters, but the discussion always focus on the grazing time and concentrate supplementation, and this is also the international conventions when studying feeding system (If all factors are considered, feeding system study is difficult to carry out). Moreover, the “Materials and Methods” and “Discussion” parts were improved based on our previous publication.

Specific comments:

L72 What is meant with “lambs synthetically”? Please reword to clarify.  

Response to the reviewer: The sentence was revised as suggested by the reviewer.

Table 1 Remove from the manuscript or indicate that it was published before. Obviously, the different feeding strategies resulted in different energy intake among the groups, which should have the main influence on the results.

Response to the reviewer: Table 1 and table 2 were the basic information of the experiment design, but published before, so we supplemented the reference in the revised manuscript.

L101 How was the pasture feed intake determined and used in later analyses? I do not see it considered in the results or discussion.

Response to the reviewer: The result of feed intake was published, so we added it in the “Materials and Methods” part and cited the reference in the present manuscript. The sentence was also revised.

L118 What was the reason for slaughter lambs using halal procedures?

Response to the reviewer: Ningxia is a Hui autonomous region, and Halal slaughter procedures is a custom of Hui nationality.

L120 Please clarify how you defined the EBW. What about life end weight, feed intake, energy intake and parameters of growth like daily weight gain? Those data should be provided or referenced, when published elsewhere. Can you show growth curves?

Response to the reviewer: Information about this part was corrected, not the EBW, but the pre-slaughter weight (PSW), and “EBW” was replaced by “PSW” in the revised manuscript. The results of final body weight, feed intake and daily weight gain were referenced in the “Materials and Methods” part. The body weight data was published, so we can’t show the growth curve in the present manuscript.

L164 correct “Shimadzu”

Response to the reviewer: The word was corrected.

L180-183 Which are the fixed factors and random factors included in the statistical model?

Response to the reviewer: The statistical analysis was corrected.

Tables: include a footnote explaining the treatment groups. Insert in footnotes the superscript letters used to indicate significant differences.

Response to the reviewer: Footnotes were revised according to the reviewer’s comments.

Table 9: superscript letter “d” in G2 for TVFC should probably be “c”. If Heterocyclic compounds are specified in a footnote, this should be done for all classes of compounds or for none.

Response to the reviewer: “d” was replaced by “c”. The footnote of heterocyclic compounds was removed.

L273 How was the value of the provided feed in this study in comparison to the nutritional requirements of lambs? Include respective discussion.

Response to the reviewer: The concentrate diet was supplemented as following (previous publication): Two-week pre-experiment was conducted before the normal experiment to determine the quantity of concentrate supplemented making sure lambs could eat up all the concentrate and to let lambs be used to the feeds. And before changing the quantity of concentrate, another two-week transitional period was needed to confirm the quantity of concentrate. The lambs were fed twice per day at 6:00 before grazing and 19:00 after grazing in individual pens. Make sure that the lambs had enough time to take all the concentrate each time.

In addition, feed intake data was cited in the revised manuscript, the value of feed was discussed in our previous publication. So in the present study, we cited the reference and had a brief discussion.

L278 Discuss the relationship between live end weight and EBW for lambs in this study.

Response to the reviewer: The discussion was supplemented in the revised manuscript.

L306-307 This would be expected with increased energy intake.

Response to the reviewer: The sentence was revised.

L328-329 Lambs of the G4 group received the same amount of concentrate as G8 lambs according to Table 1.

Response to the reviewer: The sentence was corrected.

L346-347 Clarify “restricted access time”

Response to the reviewer: The sentence was revised as “…as grazing time and the amount of concentrate supplementation changed”.

L375-378 Reword to clarify the sentence.

Response to the reviewer: The sentence was corrected.

L382 delete “were”

Response to the reviewer: “were” was deleted.

L398-400 revise this sentence

Response to the reviewer: The sentence was revised.

L402 Do you mean “more intense” or “higher”?

Response to the reviewer: The word “intense” was replaced by “higher”.

L408 replace “concentrated-based“ with “concentrate-based”

Response to the reviewer: The word was corrected.

L413 replace “feed” with “were fed” and “mechanism” with “mechanisms”

Response to the reviewer: The words were revised according to the reviewer’s comments.

L424 It is important to mention that these lambs received greater amounts of concentrates.

Response to the reviewer: The sentence was revised as suggested by the reviewer.

The Conclusion is mainly a summary and should be revised.

Response to the reviewer: We revised the conclusion part.

Round 2

Reviewer 2 Report

The authors have addressed all my comments and requests sufficiently. 

Author Response

The authors have addressed all my comments and requests sufficiently. 

Response to the reviewer: Thanks you again for helping us improve the quality of manuscript.

This manuscript is a resubmission of an earlier submission. The following is a list of the peer review reports and author responses from that submission.

Round 1

Reviewer 1 Report

I have attached most of my comments and suggested change in the attached file.  One major aspect where i felt the information provided was insufficient relates to the volatile analysis. The method was poorly described, especially in relation to the quantification but also the identification of the volatile compounds.  In addition it was unclear as to why SPME was used at such high temperatures, which i am sure resulted in the low numbers of volatiles identified.  Also it was even unclear if it was headspace SPME.  This aspect requires significant improvement.  Otherwise no other major concerns were evident.

Reviewer 2 Report

Line 90 - no ethics approval number is provided?

Line 93 - The experiment is fatally floored in design.  There were 5 treatments of which 4 were at pasture, but the animals were run in groups so there was NO replication of each treatment.  Thus the work cannot be considered for publication.